

# Vector distribution and transmission risk of the Zika virus in South and Central America

Sarah Cunze, Judith Kochmann, Lisa K. Koch, Elisa Genthner and Sven Klimpel

Goethe University, Institute of Ecology, Evolution and Diversity; Senckenberg Biodiversity and Climate Research Centre, Frankfurt am Main, Germany

## ABSTRACT

**Background:** Zika is of great medical relevance due to its rapid geographical spread in 2015 and 2016 in South America and its serious implications, for example, certain birth defects. Recent epidemics urgently require a better understanding of geographic patterns of the Zika virus transmission risk. This study aims to map the Zika virus transmission risk in South and Central America. We applied the maximum entropy approach, which is common for species distribution modelling, but is now also widely in use for estimating the geographical distribution of infectious diseases.

**Methods:** As predictor variables we used a set of variables considered to be potential drivers of both direct and indirect effects on the emergence of Zika. Specifically, we considered (a) the modelled habitat suitability for the two main vector species *Aedes aegypti* and *Ae. albopictus* as a proxy of vector species distributions; (b) temperature, as it has a great influence on virus transmission; (c) commonly called evidence consensus maps (ECM) of human Zika virus infections on a regional scale as a proxy for virus distribution; (d) ECM of human dengue virus infections and, (e) as possibly relevant socio-economic factors, population density and the gross domestic product.

**Results:** The highest values for the Zika transmission risk were modelled for the eastern coast of Brazil as well as in Central America, moderate values for the Amazon basin and low values for southern parts of South America. The following countries were modelled to be particularly affected: Brazil, Colombia, Cuba, Dominican Republic, El Salvador, Guatemala, Haiti, Honduras, Jamaica, Mexico, Puerto Rico and Venezuela. While modelled vector habitat suitability as predictor variable showed the highest contribution to the transmission risk model, temperature of the warmest quarter contributed only comparatively little. Areas with optimal temperature conditions for virus transmission overlapped only little with areas of suitable habitat conditions for the two main vector species. Instead, areas with the highest transmission risk were characterised as areas with temperatures below the optimum of the virus, but high habitat suitability modelled for the two main vector species.

**Conclusion:** Modelling approaches can help estimating the spatial and temporal dynamics of a disease. We focused on the key drivers relevant in the Zika transmission cycle (vector, pathogen, and hosts) and integrated each single component into the model. Despite the uncertainties generally associated with

Corresponding author
Sarah Cunze,
cunze@bio.uni-frankfurt.de

modelling, the approach applied in this study can be used as a tool and assist decision making and managing the spread of Zika.

# INTRODUCTION

Vector-borne diseases are globally emerging or re-emerging as a consequence of global warming, globalisation in trade and travel, changes in modern transport networks, and urbanisation (*Medlock & Leach, 2015*; *Balogun, Nok & Kita, 2016*). The arthropod-borne Zika flavivirus (ZIKV) is of great medical importance as it has the potential to cause serious complications in a small proportion of infected individuals (*WHO, 2016*; *Krauer et al., 2017*). Recent epidemics of Zika disease (or Zika fever, or Zika) in South America in 2015 and 2016 have raised awareness of an existing need for a better understanding of the geographical patterns of the ZIKV transmission risk.

Zika has not been recognised as an important disease in humans for a long time after its first discovery in 1947 in Uganda (*Županc & Petrovec, 2016*), probably due to previous outbreaks being limited in size and poor birth defect surveillance in areas where it is transmitted. Only a small proportion of cases of ZIKV infections are symptomatic (*Funk et al., 2016*). For a long time, symptomatic ZIKV infections were only reported from few small areas in Africa and Asia, and ZIKV was thus considered to be of limited public health importance until 2007 (*Lessler et al., 2016*; *Messina et al., 2016*). In 2014, ZIKV was found in the Americas for the first time, and it rapidly spread throughout several South American states. Following a considerable increase in the incidence of microcephaly in newborns, especially during an outbreak in Brazil in 2015, and of Guillain-Barré syndrome, alarms were raised worldwide (*Lessler et al., 2016*). In response to these potentially severe complications and the rapid expansion of ZIKV into previously unaffected areas, as well as the size of the outbreak, on 1 February 2016 the World Health Organization (WHO) declared a public health emergency of international concern (*Županc & Petrovec, 2016*; *Lessler et al., 2016*). Despite declining numbers of Zika infections throughout 2017 and 2018, the Pan American Health Organization (PAHO) maintains reports of cases of ZIKV infections and congenital Zika syndrome from the Americas (*WHO, 2019*).

Mapping the geographical distribution of diseases and identifying potential areas under risk of infections is crucial for evidence-based decision making in public health (*Escobar et al., 2017*). Several modelling attempts have recently been made for Zika based on correlative (*Samy et al., 2016*; *Messina et al., 2016*; *Lo & Park, 2018*) and mechanistic approaches (*Funk et al., 2016*; *Suparit, Wiratsudakul & Modchang, 2018*). Applications of correlative ENM approaches focused on modelling the potential distribution of single 'components' (cf. *Johnson, Escobar & Zambrana-Torrelio, 2019*) of the transmission cycle (e.g. *Baak-Baak et al. (2017)* for vector species and *Samy, Campbell & Peterson (2014)* for

vector species and parasites), such as pathogens (bacteria, viruses, or parasites), vectors, and hosts (*Escobar & Craft, 2016*; *Johnson, Escobar & Zambrana-Torrelio, 2019*). Other approaches directly linked disease occurrences to environmental factors (*Pigott et al., 2015*), which is especially useful when information on transmission mechanisms or distributional data of vectors or pathogens is missing. Given the large number of important factors affecting the temporal and spatial patterns of a disease and which cannot be adequately mathematically described in their complexity, the integration of empirical data in correlative approaches seems to be the most feasible way to map the geographical distribution of diseases (cf. *Messina et al., 2015*).

Here, we chose a correlative ENM approach to estimate the ZIKV transmission risk in South and Central America. Our approach is a mix between the two aforementioned approaches, accounting for the key components of the transmission cycle (e.g. modelled habitat suitability for vector species, ECM of virus distribution) and linking those to occurrences of disease cases. Reliable estimations in ENM should account for all relevant factors shaping the potential distribution of the disease. A major component of the ZIKV transmission cycle are the vector species and their distribution. ZIKV is transmitted by mosquitoes, with *Aedes aegypti* and *Ae. albopictus* being the primary and secondary vectors (*Heitmann et al., 2017*). Hence, when modelling the geographical distribution of the ZIKV transmission risk, the distribution of these two vectors needs to be established. In our approach, we first estimated the climatic habitat suitability for the two main vector species (*Ae. albopictus* and *Ae. aegypti*) to generate a proxy for the vector species' distribution. Secondly, we used regional information on disease incidences as a proxy for the presence of the virus itself. This information was based on databases of several health organisations (i.e. World Health Organization—WHO; Centers for Disease Control and Prevention—CDC; Global Infectious Disease and Epidemiology Network—Gideon) and accounted for by creating a commonly called evidence consensus map (ECM). In addition to the distribution of these key actors (vector species and virus), abiotic factors such as temperature also play an important role. We included mean temperature of the warmest quarter, as temperature is considered an important driver of the virus transmission risk for vector-borne diseases (*Mordecai et al., 2017*; *Tesla et al., 2018*). We further considered population density and gross domestic product to include socio-economic factors, potentially related to the risk of infection (cf. *Romeo-Aznar et al., 2018*).

Our aim was to describe a general pattern and test the relevance of each predictor based on current, easily accessible information using ENM as a tool. The approach applied in this study might be used as a tool and assist decision-making and managing the spread of Zika.

## MATERIAL AND METHODS

We used a correlative approach to estimate the ZIKV transmission risk in South and Central America following the approach suggested in *Messina et al. (2015)*. The graphical abstract given in Fig. 1 refers to the data used in our modified approach. The study area covers 27 countries in Central and South America.

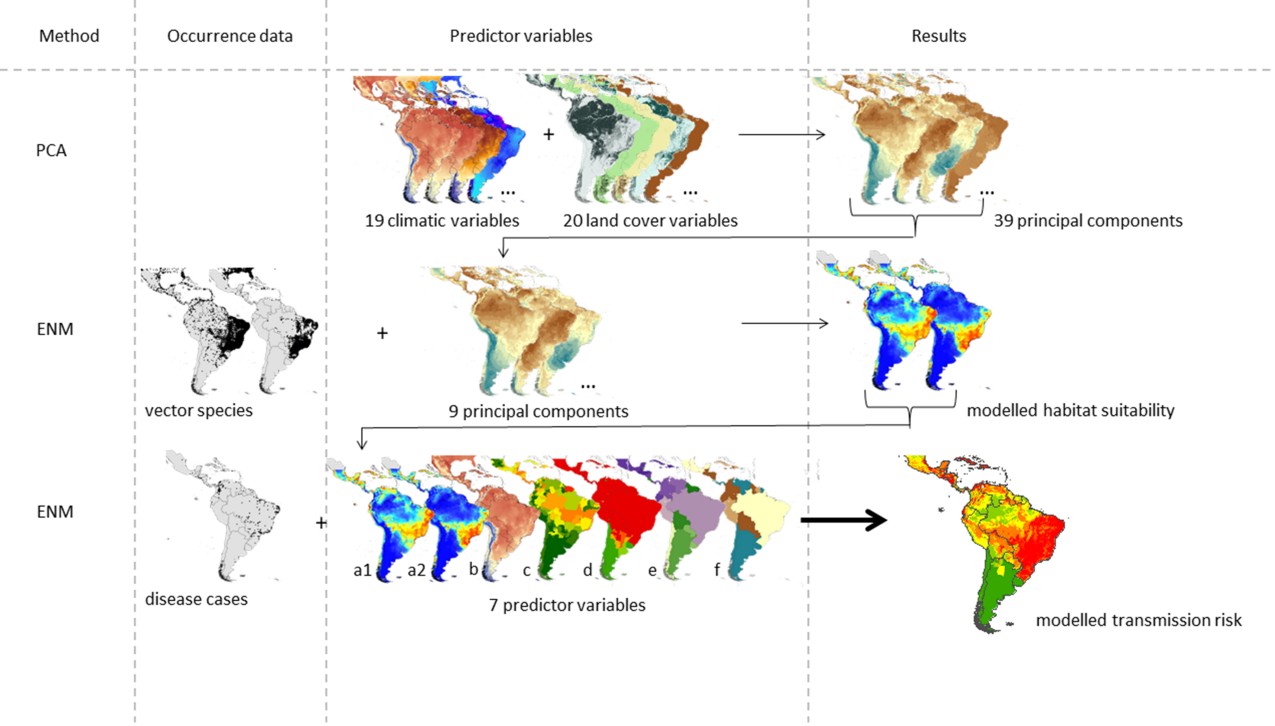

**Figure 1 Graphical abstract of the considered approach modified after *Messina et al. (2015)*.** The ZIKV transmission risk is modelled based on the following seven predictor variables: modelled habitat suitability for the two main vector species (a1) *Aedes aegypti* and (a2) *Aedes albopictus*, (b) temperature of warmest quarter, (c) Zika evidence consensus map, (d) Dengue evidence consensus map, (e) population density and (f) gross domestic product.

The final model of the ZIKV transmission risk is based on known occurrences of the disease taken from *Messina et al. (2016)* and *Messina & Shearer (2016)*, $N = 239$) and the following predictor variables: (a) habitat suitability maps of the two main vector species, *Ae. albopictus* and *Ae. aegypti* (as a proxy for vector species' distribution, see below for further details), (b) temperature of the warmest quarter, (c) ECM for Zika (cf. *Brady et al., 2012*), (d) ECM for dengue (with the same vector species as Zika), (e) population density, and (f) gross domestic product per capita (on country level).

## ENM for vector species

As species' distribution is mainly driven by climatic conditions on a continental scale, we considered 19 bioclimatic variables provided by worldclim (version 2.0; www.worldclim. org, *Fick & Hijmans, 2017*) to model the habitat suitability for the two main vector species. In addition to climatic conditions, land cover is considered to be an important driver shaping species' distributional patterns. Land cover variables were based on data from the 'GlobeCover 2009 land cover map' (*Arino et al., 2012*) provided by the European Space Agency (*ESA GlobeCover 2009 Project, 2010*). According to the United Nations Land Cover Classification Systems land cover is classified into classes, out of which we chose 20 classes for the considered study area (see Table S2). Based on the spatial resolution of the climatic data used, we calculated the percentage of each land cover class per grid cell,

resulting in 20 variable layers with the percentages of the respective land cover class ranging from 0% to 100%.

To reduce the dimensionality of predictor variables, we applied two approaches. First, we chose 10 out of the 39 original variables that we consider ecologically relevant (five climatic and five landcover variables). Secondly, we performed a principal component analysis (PCA) for the 39 variables (i.e. 19 bioclimatic variables and 20 land cover variables) using the R function "rasterPCA" implemented in the package "RStoolbox" version 0.1.10 (*Leutner & Horning, 2017*). For further analysis, we used the ENM results with the comparably highest area under the receiver operating characteristic curve (AUC) value.

To model the habitat suitability of the two main vector species *Ae. aegypti* and *Ae. albopictus* as well as the ZIVK transmission risk, we chose the maximum entropy approach implemented in the Maxent software (version 3.4.1) (*Phillips et al., 2017*; *Phillips, Dudík & Schapire, [Internet]*), which is a presence-background approach and particularly suitable when reliable absence data is missing.

Occurrence data for both species were taken from *Kraemer et al. (2015*, $N = 6{,}716$ occurrences for *Ae. aegypti* and $N = 3{,}526$ occurrences for *Ae. albopictus*). Occurrence records were adjusted to the spatial resolution of the environmental data (55 arc minutes) and only one record per grid cell was used for ENM analysis in order to reduce spatial biases (*Aiello-Lammens et al., 2015*) resulting in $N = 5{,}150$ occurrences for *Ae. aegypti* and $N = 3{,}303$ occurrences for *Ae. albopictus*.

As predictor variables we used the first nine principal components derived from the above mentioned PCA (with corresponding eigenvalues greater than one; Kaiser criterion), explaining 66.9% of the variability of the original data set of 39 variables (Fig. 1). We thus accounted for inter-correlation and were able to reduce the number of variables incorporated in the ENM with the least possible loss of information.

Two habitat suitability maps with continuous values between 0 (no habitat suitability) and 1 (full habitat suitability) for *Ae. aegypti* and *Ae. albopictus* were created.

## ENM for the ZIKV transmission risk

To model the ZIKV transmission risk we accounted for the main key drivers: vector distribution, virus distribution, temperature and socio-economic factors. Specifically, we used the following seven predictor variables: the modelled habitat suitability for *Ae. albopictus* and *Ae. aegypti* (see section 'ENM for vector species'), mean temperature of the warmest quarter (bio10) as provided by worldclim, the ECMs for Zika and dengue, as well as population density and GDP (see below for further details).

The temperature of the warmest quarter was chosen as abiotic variable due to the strong impact of temperature on virus transmission (*Mordecai et al., 2017*; *Tesla et al., 2018*). Detailed descriptions of the other variables are following below.

### ECM for Zika

The method of ECM (see *Brady et al., 2012*) has been developed to better compile disparate data types and synthesise information in a robust and repeatable way. It generally summarises information for a country or other administrative units and displays it in a map.

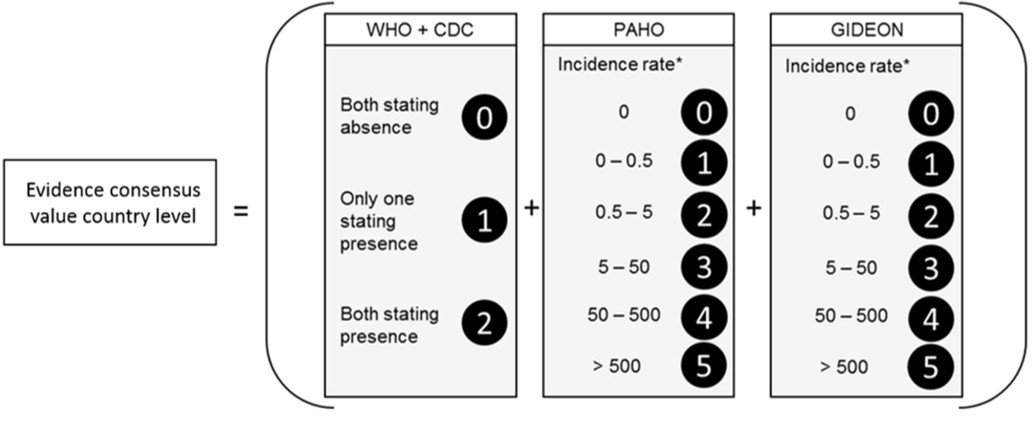

**Figure 2** **Scoring system to calculate the evidence consensus (EC) values on a country level, modified version after** *Brady et al. (2012)*. Information provided by the WHO, CDC, PAHO and GIDEON. The classification was relative to the regional values, which in turn were derived from the national class according to WHO and GIDEON.               

We compiled and standardised information on reported Zika cases between 2015 and 2017 from various sources, specifically from the World Health Organization—WHO (http://www.who.int/en/); Centres for Disease Control and Prevention—CDC (https://www.cdc.gov/); Global Infectious Disease and Epidemiology Network—Gideon (https://www.gideononline.com/), Pan American Health Organization (PAHO) (http://www.paho.org/hq/). The considered administrative units were regions, departments, provinces and states (depending on the country). The geometries of these areas were taken from the esri ArcGIS online database.

With reference to *Brady et al. (2012)* we first calculated the evidence consensus value for the Zika transmission risk. These were based on country level records obtained by the above mentioned sources, ranging from 0 (absence) to 12 (highest presence) according to the scoring system displayed in Fig. 2. In a second step, this information was refined, using more regional data on incidence rates or case numbers (depending on availability). For each administrative unit (region, department, etc.) the country level value was scaled by an administrative-unit-specific value between 0 and 1 referring to the relative incidence rate or case number in this region.

### ECM for dengue

Infections of ZIKV together with infections of DENV are known from various studies (*Dupont-Rouzeyrol et al., 2015*) and it has been suggested that the risk of symptomatic Zika is higher in people who have had prior dengue infections (*Dejnirattisai et al., 2016*; *Cohen, 2017*). We thus used the dengue ECM generated by *Brady et al. (2012)*, which ranges from

complete absence to good absence, moderate absence, poor absence, intermediate, poor presence, moderate presence, good presence and complete presence. We transformed these nine classes to numbers from 0 to 8, with only five different categories occurring in the study area (0 complete absence, two moderate absence, six moderate presence, seven good presence, eight complete presence).

### Population density and gross domestic product

Additionally, we considered the population density (population number on a country scale was taken from Gideon; information on area was taken from https://www.worldatlas.com) and the GDP per capita (*World Bank, 2016*, https://data.worldbank.org/) on a country level as predictor variables.

## Further analyses

Among the seven predictor variables for the ZIKV transmission risk, the evidence consensus for Zika as well as for dengue were considered as categorical variables. The other five variables (habitat suitability for *Ae. aegypti*, habitat suitability for *Ae. albopictus*, mean temperature of the warmest quarter, population density and GDP were considered as continuous variables. We displayed the one-variable response curves built with Maxent to assess whether the relationships between the modelled ZIKV transmission risk and any of the predictor variables were as expected. We assumed the following relationships: (a) the higher the habitat suitability for the vector species, the higher the ZIKV transmission risk, (b) the higher the temperature, up to a certain threshold, the higher the ZIKV transmission risk, (c) the higher the incidence rates of Zika (accounted for by the evidence consensus values for Zika), the higher the ZIKV transmission risk, (d) the higher the incidence rates of dengue, the higher the incidence rates of Zika (as both diseases share the same vector species), (e) the higher the population density, the higher the ZIKV transmission risk, and (f) the higher the GDP per capita, the lower the ZIKV transmission risk.

A jack-knifing test for variable importance is implemented in the Maxent software and was undertaken to analyse training gains.

## Software

The maximum entropy modelling approach implemented in the software Maxent (version 3.4.1; *Phillips et al., 2017*; *Phillips, Dudík & Schapire, [Internet]*) was used for modelling the habitat suitabilities of the two vectors as well as for the ZIKV transmission risk. We used the default settings in Maxent but only used linear, quadratic and product features, and excluded hinge features (cf. *Cunze & Tackenberg, 2015*). The restriction to linear, quadratic and product features yields smooth response curves (continuous and differentiable response curves). In addition, we enhanced the maximum number of iterations to 50,000 to ensure convergence. Maxent is an algorithm that is often used and performs well in model comparison (*Elith et al., 2006*). As a 'presence-background model', Maxent is especially suitable for studies without reliable absence data.

Further analysis was performed in *R Core Team (2014)* and ESRI ArcGIS (Release 10.3) and the latter one was also used building the maps.

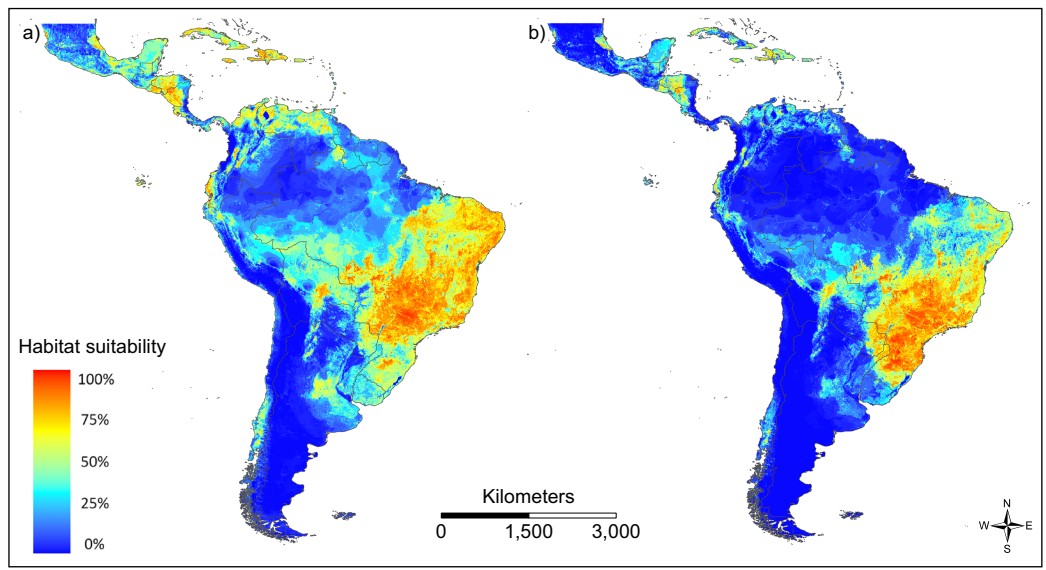

**Figure 3** Modelled habitat suitability of the two main vector species (A) *Aedes aegypti* and (B) *Aedes albopictus*. Warmer colours indicate higher modelled habitat suitability.

## RESULTS

Ecologically relevant variables were chosen based on higher AUC values, thus, we chose the PCA approach over the approach using 10 variables for further analysis. For *Ae. aegypti* the AUC values for the ENMs based on nine PCA components was AUC = 0.788 and thus, slightly higher than the AUC value of 0.732 for an ENM based on five land cover variables and five climatic variables. For *Ae. albopictus* the AUC values for the ENMs based on nine PCA components was AUC = 0.867, again slightly higher than the AUC value of 0.803 for an ENM based on five land cover variables and five climatic variables.

The patterns of modelled habitat suitability for both vector species is in good accordance with the patterns of their observed occurrence data (AUC = 0.788 for *Ae. aegypti* and AUC = 0.867 for *Ae. albopictus*; Fig. 3). Both vector species show a similar distribution, with *Ae. aegypti* showing also occurrences beyond the main distribution of *Ae. albopictus* (Fig. 3A). This could be related to the fact that this species has been spreading in South America for a longer time than *Ae. albopictus*.

We provide an ECM (sensu *Brady et al., 2012*) for Zika in South and Central America on a more regional scale (Fig. 4).

For southern parts of South America as well as the Amazon basin only very low ZIKV transmission risks ("nearly absent" or "low") were modelled (AUC = 0.860 for the ZIKA transmission risk model; Fig. 5), whereas large parts of Brazil (especially the east of Brazil), Colombia and Venezuela, Mexico and Central America as well as the Caribbean were modelled to be at a "very high" risk. The following countries are modelled to be particularly affected: Brazil, Colombia, Cuba, Dominican Republic, El Salvador, Guatemala, Haiti, Honduras, Jamaica, Mexico, Puerto Rico and Venezuela.

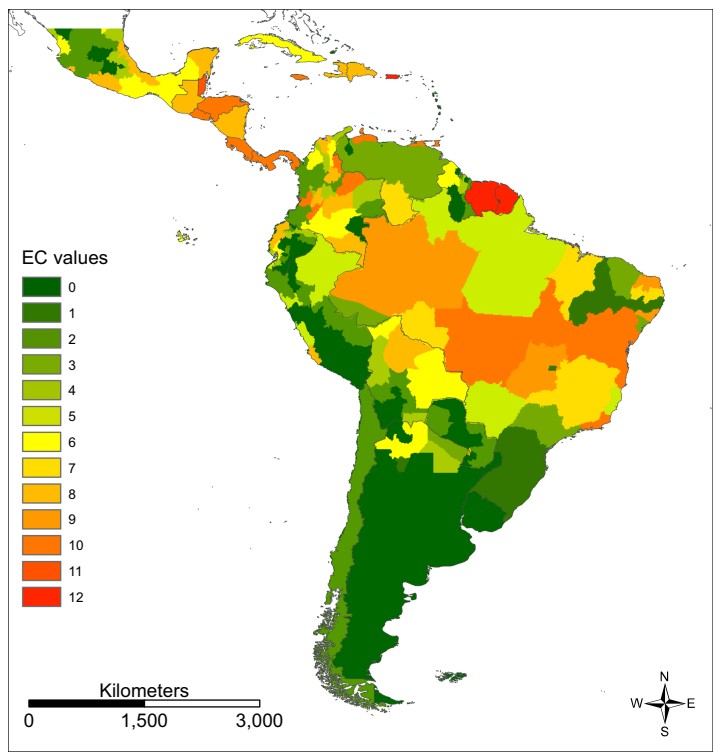

**Figure 4 Evidence consensus map for Zika on a regional scale.** Evidence consensus values between 0 (dark green) and 12 (red) according to the scoring system described in Fig. 2 for Zika in South and Central America.

According to the one-variable response curves (Fig. S1) there are positive relations between the modelled ZIKV transmission risk in South and Central America (cloglog Maxent output) and: (a) the modelled habitat suitability for *Ae. aegypti*—the main vector species for Zika in South and Central America (i.e. the higher the modelled habitat suitability for *Ae. aegypti*, the higher the modelled ZIKV transmission risk), (b) the modelled habitat suitability for *Ae. albopictus*—the second main vector species for Zika in South and Central America (here not monotonic but with an optimum at rather high modelled habitat suitabilities for the vector species, (c) the mean temperature of the warmest quarter (again, not monotonic but with an optimum at high temperatures), (d) the evidence consensus classes (categorical variable) for Zika and dengue, and e) the population density. On the other hand, the modelled ZIKV transmission risk (cloglog Maxent output) and the GDP have a monotonic negative relationship.

Results of jack-knifing (Fig. S2) for the model are displayed with only the considered variable in isolation (dark blue bars), for the model leaving out the respective variable but considering all other variables (green bars), or for the regularised training gain for the model with all seven variables (red bar).

The environmental variable with the highest gain in the jack-knifing (Fig. S2) when used in isolation is the modelled habitat suitability for *Ae. aegypti*, the main vector for ZIKV transmission. The environmental variable that decreases the model's gain the most when omitted is the Zika evidence consensus class. These classes therefore appear to have
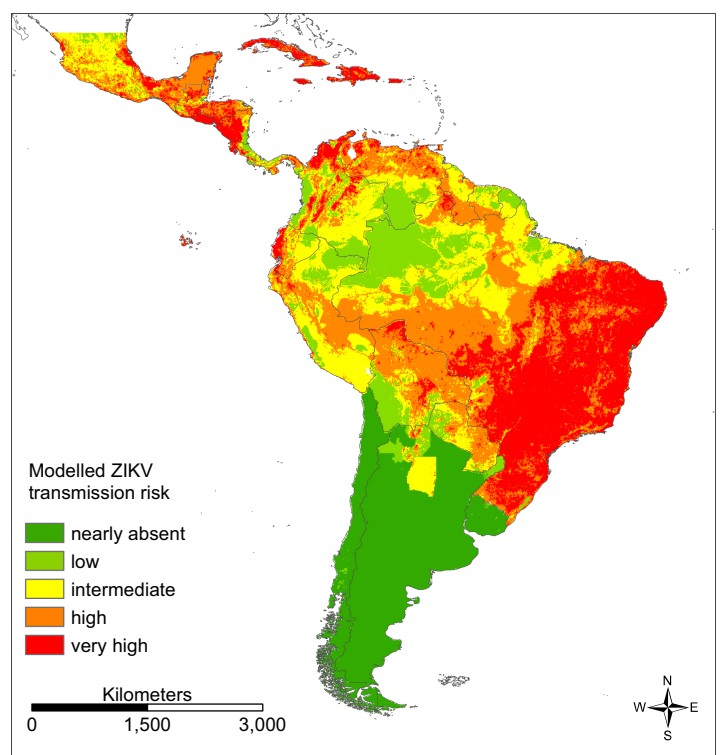

**Figure 5 Modelled ZIKV transmission risk in South and Central America based on Maxent cloglog output.** Dark green: 0–0.03, light green: 0.03–0.16, yellow: 0.16–0.30, orange: 0.30–0.62, red: 0.62–1. Classified by percentiles.

most of the information not present in the other variables. Population density and GDP show the lowest contribution to the model according to the jack-knifing, with the lowest gain if used in isolation and the lowest effect on the model gain when left out.

## DISCUSSION

The spatial and temporal patterns of vector-borne diseases are driven by a high number of different environmental, ecological and socio-economic factors with unknown influence and interaction strength (*Lessler et al., 2016*), challenging predictive models on the ZIKV transmission risk. Recently, there have been several attempts to model the ZIKV transmission risk on a global (*Messina et al., 2016*; *Samy et al., 2016*; *Carlson, Dougherty & Getz, 2016*; *Caminade et al., 2017*) or more local scale (e.g. *Wiwanitkit & Wiwanitkit, 2016* for Thailand, *Lo & Park, 2018* for Brazil), however, the spatial and temporal patterns of ZIKV transmission are still not very well understood.

Ecological niche modelling has become one of the most widely used approaches in estimating the geographical distribution of emerging infectious diseases (*Escobar & Craft, 2016*). Here, it was used to gain a better understanding of the recent spread of ZIKV in the tropical countries of South and Central America and to identify areas with a high ZIKV transmission risk. We accounted for key drivers in the transmission cycle of vector-borne disease (i.e. vector, pathogens, and hosts) and used a set of variables, which
are considered to be potential drivers of both direct and indirect effects on the emergence of Zika.

Generally, the map of the modelled ZIKV transmission risk is in large accordance with the occurrence of the mosquitoes, that is, the ZIKV transmission risk seems to be highest in the eastern coast of Brazil as well as in Central America. The relation between the single predictor variables and the modelled ZIKV transmission risk as response variable shown in the response curves matches our expectations. Results of the jack-knifing identified the vector species distribution (habitat suitability maps for *Ae. aegypti* and *Ae. albopictus*) and the Zika distribution (Zika ECM) as the most important predictor variables (as expected), whereas the population density and GDP with a very coarse spatial resolution on country scale were of minor relevance for the model. In the following paragraphs we discuss the explanatory variables used in the modelling approach in more detail.

The primary source of ZIKV infection in humans is from bites of infected mosquitoes (*Lessler et al., 2016*). We therefore took the modelled habitat suitability for the primary and secondary vectors of ZIKV transmission, *Ae. aegypti* and *Ae. albopictus* (*Heitmann et al., 2017*), respectively as a proxy for vector distribution. Both species show similar distribution patterns (based on the recorded occurrences), leading to potential competitive exclusion in certain regions (*Lounibos & Juliano, 2018*). However, in our case this does not contradict the general approach, since both species are competent vectors. Both vector species show their main distribution along the eastern coast of Brazil. This area is as a hotspot of occurrence records and consequently shows a high projected habitat suitability modelled for both species.

When modelling the habitat suitability for the two vector species, we accounted for climatic factors as well as land cover as important drivers shaping their potential distribution. The PCA was implemented to reduce the dimensionality of predictor variables (*Anderson & Gonzalez, 2011*; *Radosavljevic & Anderson, 2014*) although this implicates that modelling results are more difficult to interpret ecologically. An alternative way would have been to only choose environmental variables supposed to be ecologically relevant to the species at hand (*Anderson, 2013*). In fact, we ran two different models for each vector species (one based on the principle components and one based on five climatic and five land cover variables considered to be relevant). Both approaches yielded similar patterns of the modelled habitat suitability of the vector species. We decided to take the models showing the higher AUC values for further analyses, that is, the ENM models based on nine PCA components for both vector species.

As expected, we found a positive relationship (monotonously increasing response curve) between the modelled ZIKV transmission risk and the modelled habitat suitability for both vector species. The modelled habitat suitability for both vector species is highly important for the model, with high gain values (i.e. a quality measure for modelling, the higher the gain, the better the model) using either of the two as the only variable for the model, and a high loss in explanatory gain/importance when leaving the variable out, especially for *Ae. aegypti*. The lower effects when omitting the habitat suitability for

*Ae. albopictus* from the model can be explained by the high explanatory power of the other, ecologically similar vector species still present in the model, *Ae. aegypti*.

In addition to the distribution of a disease vector, the distribution of the virus and its prevalence are also very important factors when assessing the risk of getting infected. As an estimator for this factor, we incorporated the commonly called ECM. We provided a Zika ECM based on available information with a higher spatial resolution than only country level, which to our knowledge is something that has not been attempted before. However, the resolution still remains quite low. This is reflected in the final risk map and becomes apparent in discontinuities in the modelled ZIKV transmission risk, for example, following regional boundaries, especially in Argentina and its neighbouring countries. Our approach for the regional ECM might underestimate the occurrence of the disease for some regions. For example, for the state of Pernambucco (north-eastern Brazil), our regional ECM gives a relatively low EC value of 1.6. Although this value refers to 896 reported possible cases of Zika (*De Brito et al., 2016*), the prevalence of Zika in this region is clearly lower than in other Brazilian states according to the PAHO case reports (http://www.paho.org/data). The general patterns are well reflected by our map. The distribution and prevalence of the virus are very important factors when assessing the risk of infection. Our Zika ECM based on incidences of Zika infections is one of the best available estimators for the prevalence of the virus (provided as Ascii-file in Supplemental Information).

We used the dengue ECM as predictor variable assuming a positive correlation between the occurrence of DENV infections and the occurrence of ZIKV infections as both viruses share the same vector species. At the same time, there is some evidence that dengue immunity can be protective against Zika infection (*Gordon et al., 2019*, *Rodriguez-Barraquer et al., 2019*). In the jack-knifing, the Zika ECM clearly shows a higher variable contribution compared to the dengue ECM, with the highest loss in training gain when leaving the respective variable out. Both variables were taken as categorical variables, resulting in discontinuous response curves.

Temperature is considered a very important driver of vector-borne disease transmission due to temperature conditions affecting the life history of the vector species (*Brand & Keeling, 2017*). Mosquitoes as ectothermic insects are especially known to be temperature-sensitive. Transmission can only succeed if the mosquito is able to survive the extrinsic incubation period between becoming infectious and biting new hosts (*Tesla et al., 2018*). In addition, the transmissibility of vector-borne viruses is also supposed to be affected by temperature (*Samuel, Adelman & Myles, 2016*) as temperature conditions affect pathogen growth and survival in vector organisms (*Brand & Keeling, 2017*). It has also been suggested that temperature has an indirect effect on the vector competence of *Ae. aegypti* and *Ae. albopictus* for dengue virus transmissions through temperature dependence of *Wolbachia* infections (*Tsai et al., 2017*). To account for this assumed temperature dependency, we considered temperature in the transmission risk model, taking the temperature of the warmest quarter (bio10) provided by worldclim as a separate predictor variable. Virus transmission (Zika, dengue and chikungunya) has been suggested to occur between 18 and 34 °C with a maximum transmission rate from 26 to 29 °C

(*Mordecai et al., 2017*). The shape of the one-variable response curve for temperature dependency of the modelled transmission risk matches this suggested range very closely. We also compared areas with modelled habitat suitability for at least one of the two main vector species (in red) and the area matching the temperature criteria as suggested by *Mordecai et al. (2017)* (Fig. S4). According to these results, habitat suitability for the vector species and the optimal temperature conditions for ZIKV only match in very few areas, whereas there are wide overlaps with the broader temperature range for the ZIKV and the modelled habitat suitability for the mosquitoes. In areas with higher temperatures, that is, 26–29 °C (Fig. S4), regarded as transmission optimum according to *Mordecai et al. (2017)*, the modelled habitat suitability for the two vector species is comparatively low (Fig. 3). There are also few occurrence points for the disease (Fig. 1). The transmission risk is thus modelled to be highest in areas with temperatures below the optimum of the virus, but within the range indicated by *Mordecai et al. (2017)* and where the modelled habitat suitability for the two main vector species is high. While the modelled habitat suitability as a predictor variable shows the highest contribution to the model (Fig. S2), the temperature of warmest quarter only shows a comparatively small contribution.

As expected, the modelled ZIKV transmission risk is positively related to population density. Socio-economic factors, such as urban poverty and overcrowding, and poor public health infrastructure (cf. *Bhatt et al., 2013* considering dengue) have been suggested to impact the dynamics of ZIKV transmission. The gross domestic product per capita (GDP) was therefore included as an indirect measure of health care provision. *Gardner et al. (2018)* found regional low GDP to be the best predictor of ZIKV transmission, suggesting that Zika is primarily a disease related to poverty. However, this variable scored only a very low variable importance in the jack-knifing test, which might be explained, at least partially, by a very low spatial resolution (i.e. on country level) of population density and GDP.

Disease biogeography is a very complex field, challenging the development of reliable models for mapping disease risk. All models are therefore subject to a number of uncertainties and limitations. Crucial for the performance and reliability of models is the quality of input data, which are often biased as the reporting rate and geographical precision of occurrence data for diseases may vary greatly by country (*Messina et al., 2016*). Spatially explicit information on disease occurrences, covering both asymptomatic and symptomatic infections and derived from representative, standardised surveys would be most desirable. In practice, the location of reporting and the generally not reported site of infection can also lead to misidentification of ecological conditions favouring disease occurrence (*Allen et al., 2017*). In addition, regional differences in health care, that is, countryside vs. city, can lead to a sampling bias, with fewer cases reported from the former. Another limitation in the data used for the risk model might be that the number of reported Zika cases is likely underestimated as the majority of ZIKV infections causes only mild, flu-like symptoms (*Messina et al., 2016*), which are often unrecognised and not reported. Occurrence data affected by sampling bias is possibly the biggest weakness of any kind of modelling approach. Accounting for sampling bias in the modelling approach requires the bias being captured, for example, using information on how and where other

diseases are generally reported in South America. Without this information, a meaningful estimation of bias is not possible.

The high variability in space and time during outbreaks is typical for emerging infectious diseases like Zika. The ENM approach does not account for the temporal dynamics of the virus' occurrences. For such temporal dynamics process-based, mathematical epidemic models would be more appropriate, however, these models require many parameters, which are difficult to estimate.

The strength of the correlative approaches is that they are relatively easy to perform; however, limitations should always be acknowledged and minimised. Thus, whether introductions of ZIKV will result in its establishment and endemic disease outbreaks depends on many other factors than those considered here, for example, vector abundance, population immunity, access to health services, but also random chance (*Lessler et al., 2016*). Clearly, there is a need to continue monitoring and surveillance of all components associated to ZIKA, especially in those countries and regions where the transmission risk is high.

## CONCLUSION

Worldwide, evidence of autochthonous mosquito-borne ZIKV transmission has been reported from 87 countries and territories (as of July 2019, *WHO, 2019*) and there is still a risk for further spread of ZIKV (*WHO, 2019*). In response to the emergence and global spread of ZIKV infections and associated complications, public health systems need to be strengthened and should include epidemiological surveillance. Disease biogeography is currently a promising field to complement epidemiology. Applying the concepts and tools from ENM to disease biogeography and epidemiology will provide biologically sound and analytically robust descriptive and predictive analyses of disease distributions, help designing evidence-based control strategies, and allow comprehensive identification of potential transmission areas to allocate resources for surveillance (*Escobar et al., 2017*; *Johnson, Escobar & Zambrana-Torrelio, 2019*).

## LIST OF ACRONYMS

| | |
|---|---|
| **Ae.** | *Aedes* |
| **CDC** | Centers for Disease Control and Prevention |
| **cf.** | lat. confer |
| **ECM** | evidence consensus map |
| **ENM** | ecological niche mode |
| **ESA** | European Space Agency |
| **esri** | Environmental Systems Research Institute |
| **GDP** | gross domestic product |
| **Gideon** | Global Infectious Disease and Epidemiology Network |
| **GIS** | geographical information system |
| **LCCS** | United Nations Land Cover Classification Systems |
| **PAHO** | Pan American Health Organization |
| **PCA** | principal component analysis |

| WHO | world health organization |
|-----|---------------------------|
| ZIKV | Zika virus |

### Funding

The present study was supported by the Uniscientia Stiftung. The present study was also supported by the LOEWE-Centre TBG funded by the Hessen State Ministry of Higher Education, Research, and the Arts (HMWK). The funders had no role in study design, data collection and analysis, decision to publish, or preparation of the manuscript.

### Grant Disclosures

The following grant information was disclosed by the authors:
Uniscientia Stiftung.
Hessen State Ministry of Higher Education, Research, and the Arts (HMWK).

### Competing Interests

The authors declare that they have no competing interests.

### Author Contributions

- Sarah Cunze conceived and designed the experiments, performed the experiments, analysed the data, contributed reagents/materials/analysis tools, prepared figures and/or tables, authored or reviewed drafts of the paper, approved the final draft.
- Judith Kochmann conceived and designed the experiments, authored or reviewed drafts of the paper, approved the final draft.
- Lisa K. Koch conceived and designed the experiments, contributed reagents/materials/analysis tools, authored or reviewed drafts of the paper, approved the final draft.
- Elisa Genthner performed the experiments, analysed the data, contributed reagents/materials/analysis tools, prepared figures and/or tables, authored or reviewed drafts of the paper, approved the final draft.
- Sven Klimpel conceived and designed the experiments, authored or reviewed drafts of the paper, approved the final draft.

### Data Availability

All data used in this study is based on published data and is available online.
The Occurrence data for the vector species is available at http://dx.doi.org/10.5061/dryad.47v3c. Climate data is available at http://worldclim.org/version2. Land cover data is available at http://due.esrin.esa.int/page_globcover.php. Occurrences of Zika virus in humans is available at https://doi.org/10.6084/m9.figshare.2573629.v1. The temperature of warmest quarter is available at http://worldclim.org/version2. A Zika evidence consensus map was compiled according to Fig. 2, and is available as a Supplemental File. The dengue

evidence consensus map is available at https://journals.plos.org/plosntds/article?id=10.1371/journal.pntd.0001760. Gross domestic product per capita data is available at https://data.worldbank.org/. Population density data is available at https://www.gideononline.com/ and https://www.worldatlas.com. These links are also available in Table S1.

## Supplemental Information

Supplemental information for this article can be found online at http://dx.doi.org/10.7717/peerj.7920#supplemental-information.

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
