# Peer review of "Vector distribution and transmission risk of the Zika virus in South and Central America"

_PeerJ, doi:10.7717/peerj.7920_

## Round 0.1 · original submission · Major Revisions

Dear Dr. Conze and colleagues:

Thanks for submitting your manuscript to PeerJ. I have now received three independent reviews of your work, and as you will see, the reviewers raised some major concerns about the research. Despite this, these reviewers are optimistic about your work and the potential impact it will have on research studying ecology-based modeling of virus transmission. Thus, I encourage you to revise your manuscript, accordingly, taking into account all of the concerns raised by all reviewers.

There are many aspects of your work that stand to be improved, as outlined by the reviewers. Please consider all of the comments about your experimental design, and how it can be improved upon. There seem to be many missing references, and areas where the writing can be reworked. Above all, make sure your modelling design and implementation is thoroughly explained, such that it could be repeated by independent workers. Use the supplemental material if you need to take a lot of space with this.

Therefore, I am recommending that you revise your manuscript, accordingly, taking into account all of the issues raised by the reviewers. I look forward to seeing your revision, and thanks again for submitting your work to PeerJ.

Good luck with your revision,

-joe

·

Basic reporting

No comment.

Experimental design

No comment.

Validity of the findings

No comment.

Additional comments

The authors address an interesting question regarding the potential to predict Zika virus transmission risk in South and Central America. Given its social and health impacts, the ability to understand and manage ZIKAV transmission is an important knowledge gap. However, the authors do not present the methods in such a way that they can be sufficiently understood and reviewed. There is very little information provided about the model benefits and parameterization. How well did the models perform? The lack of detail additionally prevents robust comparison between different areas at greater or less risk of spread of the ZIKAV infection. A clear, consistent and comparable thresholding process is critical to advancing our understanding of risk. Also, there are lots of issues (grammatical and collocation of words) in some areas of the paper.
Abstract:
Compared to the Background and Methods, Results and Conclusion are too short.
Introduction:
In the last paragraph finish Introduction with possible implications of your study.
Materials and Methods:
You need to clarify why you have selected MaxEnt for the modeling, based on what kind of benefits. Also, clarify the setting you have used to perform the MaxEnt model.
Line 90: I’m not sure whether it matches with the citation guideline, but I guess it is better to move out the authors from the parenthesis: “following the approach suggested in Messina et al. (2015)”.
Line 93: specify how many points were obtained from the references and what kind of pre-processing did you perform to prepare them for the ENM analysis.
Line 103: change ‘worldclim’ to ‘WorldClim’.
Line 103: specify the resolution of the climatic variables used for ENM.
Line 106: Correct quotation at the beginning of „GlobCover 2009 land cover map”.
Line 115: change ‘packages’ to ‘package’.
Line 122: put the name of authors out of the parenthesis.
Line 122: specify how many points were obtained from the references and how many remained after the spatial screening.
Line 124: change ‘was accounted for’ to ‘was used for ENM analysis’.
Line 133-134: repeated usage of ‘account for’! for example in line 134 change ‘we accounted for’ to ‘we used’.
Line 149: remove comma after (https://www.cdc.gov/).
Results
In the Results you haven’t mentioned the predictive performance of your models at all!! At the very least, you need to report the AUC of the performed models.
You have scrimped the geographic pattern of ZIKV transmission risk to some general addressing. You need to report high-risk areas more specifically. For example, you can classify the risk maps to four equal interval classes (e.g. no-risk, low-risk, moderate risk, high-risk), calculate areas of each class and report this area at a country scale. By doing so you can specify countries with highest areas of suitability, i.e. ZIKV transmission risk, more narratively.
Line 204-205: report the AUC of the models for vector species.
Line 212-214: the ECM approach is not a modeling procedure. Replace ‘were modelled’ with ‘were assigned’.
Line 297: This citation is not listed in the References list. Besides, you don’t need to mention the complete name of the reference. You just need to refer to its authors in the main text.
Line 329-330: ‘which is a major point of criticism of correlative approaches in disease modelling overall’ this sentence needs a reference.

Reviewer 2 ·

Basic reporting

The citations given for a lot of statements tend to be from reviews rather than citing the original primary evidence.

The discussion section has a lot of repetition of the method of results of the paper, while limitation focus on more general limitations of data or classes of models rather than being specific to their analysis.

Introduction: missing a section on previous work and what specific modelling advance is being made in this paper.

Experimental design

No measures of absolute predictive performance of the aedes or Zika models are given. This is key and necessary bit of information for presentation of any predictive model.

Line 187: “the higher the incidence rates of dengue, the higher the incidence rates of Zika (due to co-infections)” – I think co-infections is probably the wrong focus here. I think the more convincing argument about why they would co-occur is that they share the vector and probably similar risk factors. There is also some evidence that dengue can be protective against Zika infection (e.g. Rodriguez-Barraquer et al. Science 2019), so this would argue against imposing a monotonic constraint on this variable.

Line 162: what was the date range for evidence included in the ECM analysis?

Line 125: why the first 9?

Line 114: give some more methodological details about how “rasterPCA” performs PCA

One major criticism of this work is the lack of detail given on the modelling approach. The authors appear to have used the MaxEnt software without any customization to this specific application. The statement that it has previous been used for other species distribution modelling examples is not sufficient given the unique nature of the Zika datasets which were collected in an international public health emergency and I cannot see any adjustment of the datasets used to consider their specific nuances in this analysis. A big concern here would be the highly variable reporting practices for Zika that vary across space and time (particularly early in the outbreak). No attempt has been made to adjust the model to account for this or assess sensitivity to such biases. I couldn’t find any information on how background points were generated- this is key in assessing how observation bias was included.

Validity of the findings

Line 313: “the importance of temperature for the ZIKV transmission risk is very low” – given temperature variables also went into the aegypti and albopictus models it is likely that these covariates soak up the variance that would alternatively have been explained by temperature, therefore I’m not sure this inference is correct.

Figure 4: the Brazilian state of Pernambucco has a low EC value, despite reporting many Zika and microcepahly cases (see Pubmid ID: 27812648).

Line 204: “good accordance with the patterns of their observed occurrence data” – need to give some statistics to justify this, a visual comparison is inadequate – AUC?

Additional comments

Figure 5: why map truncated in N. Mexico?

Figure 3: can’t see the risk maps with the dark points- maybe put on a separate map or use just circle outlines? Plotting background points would also be useful to see what the model is trying to discriminate between.

Line 41: “tourism” -> “travel” – seems unfair to single out tourism

Line 49: “As symptomatic ZIKV infections have only occurred rarely” – strange phrase, do the authors mean only a small proportion of cases are symptomatic – if so state how much and give reference(s). I would also disagree that this is the reason why it has been “of limited public health importance” – this is more due to previous outbreaks being limited in size and poor birth defect surveillance in areas where it is transmitted.

Lien 55: citation after “recongnized.”

Reviewer 3 ·

Basic reporting

- L45-46: In the description of Zika disease (lines 45-46) it is unclear what the authors mean by “short Zika”. Is “short Zika” a synonym for Zika fever/disease or do they mean “Zika” is a short-hand term for the disease?

- L49-54: I understand the general idea is clear, however these sentences could be condensed and streamlined so they are not redundant.
- L54: This is a styling preference. I would suggest substituting “But” with “However,” or a similar word.

- L64-65: Since ecological niche models are tools, I would suggest removing “tools of”.

- L66: Mismatch between noun and verb. Should read as “diseases are present”

- L67: Mismatch between noun (diseases) and possessive adjective (its). If the term is kept plural should read as “their potential distributions”

- L67: There are plenty of references that could be included here, please add some more. I would suggest looking up work developed by AT Peterson, DM Pigott, J Blackburn, among others.
- L70: Same comment as L67 regarding references

- L74: Since the authors have already introduced the abbreviation for ecological niche models (ENM) I would suggest using it in lieu of the full term.

- L81: This is a styling preference, however, the word “so-called” has two meanings which could lead to misinterpretations. I would suggest changing this word to something like “commonly called” to avoid confusion.

- L106: There is a typo, quotations before GlobCover seem to be inverted.

- L120-121: The software note corresponding to the Maxent version implemented should be cited. “Steven J. Phillips, Robert P. Anderson, Miroslav Dudík, Robert E. Schapire, Mary Blair. 2017. Opening the black box: an open-source release of Maxent. In Ecography”

- L137-138: The sources for population density and GDP data are not referenced.

- L149: Typo. There is an extra comma after the CDC website citation
- Overall inconsistencies in the way references are cited in line. For example, sometimes the authors will cite a reference as “Following methods developed by Smith et al. (year)….”, while other times they will cite “Following methods developed by (Smith et al. year)”.

- L166-168: I think it would be very beneficial for the reader if the ECM classes proposed by Brad et al. (2012) and assigned numbers were presented in a table.

Experimental design

- L81-86: It is not clear how the temperature variables mentioned were included in the analysis. I would assume this variable is already incorporated in the ENMs of both vector species, however, from the wording it seems like temperature was accounted for after these models were built. Likewise, clarification is needed to explain how these socioeconomic factors were related to risk of infection and if this is another step in the modeling.

- L90: Please add a brief explanation on why that particular methodology was chosen.
- L96-99: While I think it is wonderful that the authors include socio-economic variables to estimate risk of transmission, it is unclear why temperature of the warmest quarter is used to estimate risk. It seems like this variable would already be included in the vector distribution models. If there is a reason to do so, please provide a more elaborate explanation.

- L109: I think it would be beneficial to the reader if the authors could include a table of which land cover classes were selected for the study as well as a brief explanation as to why they were.

- L101-112: Though it is not explicitly stated, I understand the authors implemented a PCA to reduce model overfitting by reducing the dimensionality of predictor variables (Anderson & Gonzalez (2011) Ecol. Model. 222: 2796– 2811; Radosavljevic & Anderson (2014). J. Biogeo. 41(4), 629-643). However, this must be done with caution as results derived from a model using PCs are more difficult to interpret ecologically. An alternative way to deal with this problem is by only choosing environmental variables known to be ecologically relevant to the species at hand. This topic has been discussed at length by several authors (for example, Anderson (2013) Annals of the New York Academy of Sciences). The number of variables can be further reduced by using the categorical land cover raster instead of the 20 derived percentage land cover class layers that the authors used. Maxent can handle combinations of categorical and continuous variables, as the authors are aware since they use both types of variables to build the ZIKV models. Therefore, I am slightly confused as to why they didn’t use the same approach for the vector models.

- L122: The total number of mosquito occurrence records used for the vector models is not reported, nor are the total number of background/pseudoabsence points. Likewise, the authors fail to mention how the study region was delimited (i.e. from which background points will be selected for model calibration). The inclusion of areas where the species is absent despite being environmentally suitable for the species could lead to biased modeling results (Barve et al. (2011) Ecol. Model., Anderson & Raza (2013) J. Biogeog.). This is why many studies choose to delimit the study region using methods as the minimum convex hull.

- L132-142: I find the implementation of ENM to model transmission risk of infectious diseases very interesting, however, I believe justification as to why the authors believe this is an appropriate method to do so is warranted. Though I agree that mean temperature of the warmest month is likely very relevant for mosquito-borne disease transmission, I can’t help but ask whether this is because of the variable’s effect on vector biology (and was included in the vector models as well). Does temperature affect virus distribution directly or indirectly? See Anderson (2013) Annals NYAS.

- L124: I imagine the reason why only one record per grid cell was used for model construction is to reduce spatial biases, however this is not explicitly stated (See Aiello-Lammens et al. 2015 Ecograhy). Likewise, given the biology of mosquitoes and the nature of disease transmission, it is unclear why the authors chose a spatial resolution of 5 arc minutes when finer resolutions are available in Worldclim.

- L132-141: Same as L132-142

- L182-190: I appreciate that the authors attempted to lay out their hypothesized relationships between ZIKV transmission risk and each predictor variable. However, I would have liked to see the reasoning behind these assumptions to be a little more fleshed out.

- L194-199: Many studies have addressed the issue of using default Maxent settings since modeling results may be greatly affected by the selection of regularization multipliers that penalize model complexity (See Merow et al. (2013) Ecography; Elith et al. (2010) Methods Ecol. Evol). Methods for model tuning to select appropriate regularization multipliers and feature classes that optimize balance between goodness-of-fit and model complexity (i.e. model tuning) have been proposed and are widely available (See Muscarella et al. (2014) Methods Ecol Evol) and should by now be common-practice in any implementation of ENM. Furthermore, model performance can be affected by the partitioning method used in model calibration, particularly when dealing with large geographic extents (see Radosavljevic, A., & Anderson, R. P. (2014) J. Biogeog).

Validity of the findings

- L215-225: Interpretation of response curves may be a little tricky. I noticed that most of the response curves were truncated which generally means that the background was not sampled enough to actually give you a unimodal response.

- L223-236: Please be mindful that the results of variable jack-knifing only reveals which variables contributed to the model more heavily. Ecological interpretations of these results must be done with extreme caution.

- L203-236: Metrics of model performance (e.g. AUC, AICc, Omission rates) were not reported.

Additional comments

This article addresses a very relevant and current issue and proposes an interesting approach to modeling the risk of disease transmission via ecological niche modeling.

Unfortunately, there are many fundamental flaws in the methodology that must be addressed before considering this article for publication.

I do believe this article has great potential and have strived to provide comments that will be helpful in development of a revised version.

Best of luck.

---

## Round 0.2 · Major Revisions

Dear Dr. Cunze and colleagues:

Thanks for revising your work. I was only able to secure one re-review of your revision, and the reviewer is now recommending rejection. I would like you to consider the reviewer’s comments and perhaps revise your work again based on these concerns. Alternatively, you may withdraw your manuscript from PeerJ. I do think that addressing the reviewer’s concerns will greatly improve your manuscript, and thus encourage you to do so.

Good luck with your revision,

-joe

Reviewer 2 ·

Basic reporting

-

Experimental design

Original comment:
One major criticism of this work is the lack of detail given on the modelling approach. The authors appear to have used the MaxEnt software without any customization to this specific application. The statement that it has previous been used for other species distribution modelling examples is not sufficient given the unique nature of the Zika datasets which were collected in an international public health emergency and I cannot see any adjustment of the datasets used to consider their specific nuances in this analysis. A big concern here would be the highly variable reporting practices for Zika that vary across space and time (particularly early in the outbreak). No attempt has been made to adjust the model to account for this or assess sensitivity to such biases.
I couldn’t find any information on how background points were generated- this is key in assessing how observation bias was included.

Response:
We now provide more details on the specific settings we used (paragraph “Software” L226). Briefly, Maxent was used with the default settings using only LQP features to get smoother and continuously differentiable niche functions and to avoid overfitting. Background points were sampled according to the default settings.
The high variability in space and time during outbreaks is typical for emerging infectious diseases like Zika and we certainly agree that there is likely sampling bias in the data. We added another few sentences on this aspect in the discussion (L403ff). We also added a paragraph on previous modelling approaches in the Introduction (L75ff).

Re-review comment:
I see the inability or unwillingness of the authors to assess the sensitivity of their findings to their chosen method, analysis steps and parameterisations as a fatal flaw in what is primarily a modelling paper. All three reviewers have highlighted a range of analysis steps that could substantially alter the findings of these studies and I do not believe any of these have been adequately addressed by new analyses in this resubmission (particularly the comments by reviewer # 3 which contains some responses that are just plain incorrect (see reviewer 3 validity of findings section)).

The AUCs (now visible in this resubmission) are adequate, but really should be higher given the high ratio of presence : randomly sampled background points. This strongly suggests various steps of model building, fitting and evaluation need to be revisited to optimize predictive performance. I also would still like to see how AUC (as well as the rider ROC curve if the authors insist on deviating from threshold based methods) changes based on different methods of background point sampling given the, acknowledged, sampling bias of the data.

Validity of the findings

-

Additional comments

-

---

## Round 0.3 · accepted · Accept

Dear Dr. Cunze and colleagues:

Thanks for re-submitting your revised manuscript to PeerJ, and for addressing the concerns raised by the reviewer. I now believe that your manuscript is suitable for publication. Congratulations! I look forward to seeing this work in print, and I anticipate it being an important resource for research studying ecology-based modeling of virus transmission.

Thanks again for choosing PeerJ to publish such important work.

-Joe

·

Basic reporting

The authors have addressed all the comments that were notified before.

Experimental design

The authors have addressed all the comments that were notified before.

Validity of the findings

The authors have addressed all the comments that were notified before.

Additional comments

The authors have addressed all the comments that were notified before.